# Structure of the Newcastle Disease Virus L protein in complex with tetrameric phosphoprotein

Jingyuan Cong[1,2,4], Xiaoying Feng[1,2,4], Huiling Kang[3], Wangjun Fu[1,2], Lei Wang[1,2], Chenlong Wang[3], Xuemei Li[1] ✉, Yutao Chen [1] ✉ & Zihe Rao[1,3]

Newcastle disease virus (NDV) belongs to Paramyxoviridae, which contains lethal human and animal pathogens. NDV RNA genome is replicated and transcribed by a multifunctional 250 kDa RNA-dependent RNA polymerase (L protein). To date, high-resolution structure of NDV L protein complexed with P protein remains to be elucidated, limiting our understanding of the molecular mechanisms of Paramyxoviridae replication/transcription. Here, we used cryo-EM and enzymatic assays to investigate the structure-function relationship of L-P complex. We found that C-terminal of CD-MTase-CTD module of the atomic-resolution L-P complex conformationally rearranges, and the priming/intrusion loops are likely in RNA elongation conformations different from previous structures. The P protein adopts a unique tetrameric organization and interacts with L protein. Our findings indicate that NDV L-P complex represents elongation state distinct from previous structures. Our work greatly advances the understanding of Paramyxoviridae RNA synthesis, revealing how initiation/elongation alternates, providing clues for identifying therapeutic targets against Paramyxoviridae.

The Paramyxoviridae contains some of the most devastating human and animal pathogens, including measles virus (MeV), mumps virus (MuV), parainfluenza virus 5 (PIV5), Nipah virus (NiV) and NDV[1–5]. NDV causes Newcastle disease which is a highly lethal infectious disease that affects avian species, causing considerable losses for poultry industry worldwide[1,6]. NDV also infects humans, causing conjunctivitis. Despite its high pathogenicity in poultry, NDV is a promising vaccine vector against human and veterinary pathogens[7–9]. Recently, NDV was shown to exhibit oncolytic activity against many tumor types, thus constituting a potential therapeutic anti-cancer tool[10–13].

NDV is a typical non-segmented negative-sense RNA virus (nsNS RNA virus) that contains an RNA genome of 15.2k nts[14] encoding six essential genes: nucleocapsid (N), matrix protein (M), phosphoprotein (P), fusion protein (F), haemagglutinin-neuraminidase protein (HN), and large polymerase protein (L)[15–17]. The proteins N, P and L are essential for NDV genome replication and transcription during viral infection[1].

NDV L protein consists of 2204 amino acids with a molecular weight of ~250 kDa[18]. It is the last gene transcribed during the viral replication cycle. Both its transcription and replication activities critically depend on the presence of a cofactor called P protein, which assists L during RNA synthesis. The L protein consists of the N-terminal RNA-dependent RNA polymerase domain (RdRp), polyribonucleotidyl transferase domain (PRNTase), the connector domain (CD), the methyltransferase domain (MT), and C-terminal domain (CTD)[19–21]. An anti-genome copy of the RNA (cRNA) is generated during replication. This cRNA serves as the template for synthesizing full-length viral genomic RNA (vRNA). In addition, during transcription the L protein also performs 5′ capping, methylation, and polyadenylation on the newly synthesized viral mRNA[22].

[1]National Laboratory of Biomacromolecules, Institute of Biophysics, Chinese Academy of Sciences, Beijing, China. [2]University of Chinese Academy of Sciences, Beijing, China. [3]Laboratory of Structural Biology, School of Medicine, Tsinghua University, Beijing, China. [4]These authors contributed equally: Jingyuan Cong, Xiaoying Feng. ✉e-mail: lixm@ibp.ac.cn; chenyutao@ibp.ac.cn

During viral transcription and replication, the viral RNA is enveloped and protected by N proteins to form an RNA ribonucleoprotein complex (RNP). In these processes, the P protein serves a crucial role as a structural bridge that connects the L and N proteins. P protein consists of three domains: the N-terminal domain (NTD), central oligomerization domain (OD), and C-terminal X domain (XD)[23–25]. The N terminus of P forms complexes with nascent N protein monomer (N[0]) to prevent its random encapsidation of non-specific RNA[26,27]. The central region of P stabilizes the L protein in the L-P complex and is essential for connecting the L protein and the viral RNA template. The P-XD interacts with the RNPs to assist the template RNA with entering the template entry channel, and assists NTPs with entering the NTP entry channel[28]. Although in vitro studies have shown that the L protein can perform RNA polymerase activity in the absence of the P protein, polymerase activity was significantly increased in the presence of P protein[20]. Previously solved structures and functional studies of P protein established that P protein directs L protein to the RNP template. During this process, P protein assists the release of N protein from RNP, and naked RNA enters L protein for replication and transcription.

To date, several complex structures of L-P proteins from human respiratory syncytial virus (hRSV)[29,30], human metapneumovirus (hMPV)[28], PIV5[5], rabies virus (RABV)[31], Ebola virus (EBOV)[32] and vesicular stomatitis virus (VSV)[19,33] have been resolved. However, only hRSV, hMPV, EBOV as well as PIV5 contain tetrameric P proteins, while their overall structures are similar. In contrast, L-P complexes of RABV and VSV contain a short peptide derived from P proteins. However, high-resolution structures of full-length L protein in complex with P protein tetramer remain to be solved. The structures of hRSV, hMPV and EBOV L-P complexes were previously solved at high-resolution to illuminate L-P interface[28–30]; however, the CD-MTase-CTD modules of L proteins have not been solved due to their flexibility. The structure of the PIV5 L-P complex contains the full-length L protein, but the region containing the interaction interface for L protein and tetrameric P protein remains absent due to its flexible nature. Therefore, it remains unclear how the activity of the enzymatic L protein is affected by interaction with P protein. Critically, the mechanisms responsible for converting from RNA initiation to elongation state remain to be elucidated.

In summary, we report here the atomic resolution cryo-EM structures of the NDV L-P complex. The C-terminal of the L protein adopts a unique arrangement and interaction mode with the P protein tetramer.

## Results

### Cryo-EM determination of the NDV L-P complex

To determine the complex structure, we co-expressed NDV L and P proteins in *sf9* insect cells using a bac-to-bac expression system. As shown in Fig. 1a, the protein complex was isolated at high purity (>95%) and without visible degradation as assessed by SDS-PAGE. To verify that the L-P complex was functional, we measured its in vitro RNA template-dependent polymerase assay, which showed that our NDV L-P complex was catalytically active (Fig. 1b). To prepare cryo-EM samples, the purified L-P complex was plunge-frozen on amorphous alloy film (R1.2/1.3, Au, 300 mesh) at concentration of 1 mg/ml. For cryo-EM data collection, we used a 300 kV FEI Titan Krios transmission electron microscope with a GIF-Quantum energy filter (Gatan) and a Gatan K2-summit detector (Fig. 1c, d). After motion correction, CTF estimation, 2D classification, 3D classification and 3D refinement, we obtained L-P complexes in three integrity states: the full-length L-P complex (L$_f$-P), core L-P complex (L$_c$-P) and core L-extended P complex (L$_c$-P$_e$). A total of 42.5k particles remained in the L$_f$-P complex, 455k particles remained in the L$_c$-P complex and 153k particles remained in the L$_c$-P$_e$ complex. The final resolution of L$_f$-P complex, the L$_c$-P complex and L$_c$-P$_e$ were 3.41 Å, 3.0 Å and 3.25 Å respectively (Fig. 1e, f and

Supplementary Fig. 1). Statistics for cryo-EM data collection are summarized in Supplementary Table 1.

After model building and optimization, we validated the final structures using the programs CCP4[34] and PHENIX[35]. Our structural analysis showed that the L$_f$-P complex contains the RdRp domain, PRNTase domain, CD domain, MTase domain, and CTD domain of L protein and OD/XD regions of P protein forming a tetramer (Fig. 2 and Supplementary Fig. 2). In comparison, the structure of the L$_c$-P complex contains the RdRp domain and PRNTase domain of the L protein and P tetramer. In both structures, several gaps within the connecting loops were not modeled due to cryo-EM density map ambiguity in these regions (Supplementary Table 2). We also solved an L$_c$-P$_e$ complex from which we built an extended region of the P-OD tetramer (Supplementary Fig. 3). Within the structure of this complex, we identified the OD (259-302) and XD (302-399) regions of the P proteins. We then modeled these regions in the cryo-EM density map to obtain the structure of L$_c$-P$_e$ (Supplementary Table 2).

### Overall structure of NDV L protein

As shown in Fig. 2, the final atomic model of our L$_f$-P complex includes the RdRp domain, PRNTase domain, MTase domain, CD domain and CTD domain of the NDV L protein, the OD and XD domains of P proteins in tetrameric form. The RdRp domain of L protein contains a typical "fingers-palm-thumb" right-hand fold and motifs, which is highly conserved within RNA polymerases (Fig. 3a, b). The PRNTase domain participates in capping viral mRNA. This domain includes a priming loop (1186–1216) and an intrusion loop (1257–1289) located near the RdRp active site (Fig. 3c). The CD domain connects the PRNTase domain with the MTase-CTD module, rendering the structure of the L protein highly flexible. The MTase domain also harbors catalytic K-D-K-E motif (Supplementary Fig. 2), demonstrating that the MTase domain of NDV is highly conserved in Paramyxoviridae[18].

### Overall structure of the NDV P protein

The interface between L and P tetramers within the NDV L-P complex was solved at a high resolution. Our structural analysis showed that the NDV P protein is present in a unique tetrameric conformation, which is different from the conformation found in hRSV, EBOV and hMPV L-P complexes[28–30]. The four monomers of the P proteins adopt different conformations, and interact extensively with the RdRp domain of L (Fig. 4a, b). We divided the tetrameric P into four P monomers (P1, P2, P3, and P4 corresponding to chain B, C, D, E in the final PDB entries) according to the structure of the hMPV L-P complex[28]. When we analyzed the L$_c$-P structure, we found that the solved regions of each P monomer are P1 (261–301), P2 (250–305), P3 (259–312), and P4 (274–399), whereby the P4 monomer contains the largest number of residues built into the cryo-EM density map. Our structural analysis showed that the P-OD regions of the four monomers form a four-helix bundle. Due to the flexibility of P-XD, this region was only modeled in P4, where it interacts with L (Fig. 4c, d, Supplementary Fig. 4). Analysis of the complex structures showed that P1&P2 (P-OD regions, 261–289) interact with L, while P3 (260–312) and P4 (274–283, 308–315) are bound to the helix of P1&P2. Based on these findings, we built a more intact four-helix bundle of P-OD into the model of L$_c$-P$_e$. The other built region of P proteins was of similar length to its corresponding region present in the L$_c$-P complex structure (Supplementary Fig. 3).

### Conserved features of the NDV L protein RdRp active site

The RdRp region of NDV L protein contains four subdomains (palm, fingers, thumb and a structural support subdomain) and six conserved motifs (motif A-F). Five of the six motifs (A-E) are located in the palm subdomain, while motif F is located in the finger subdomain and motif E connects the palm-thumb subdomains (Fig. 3a, b). This structural

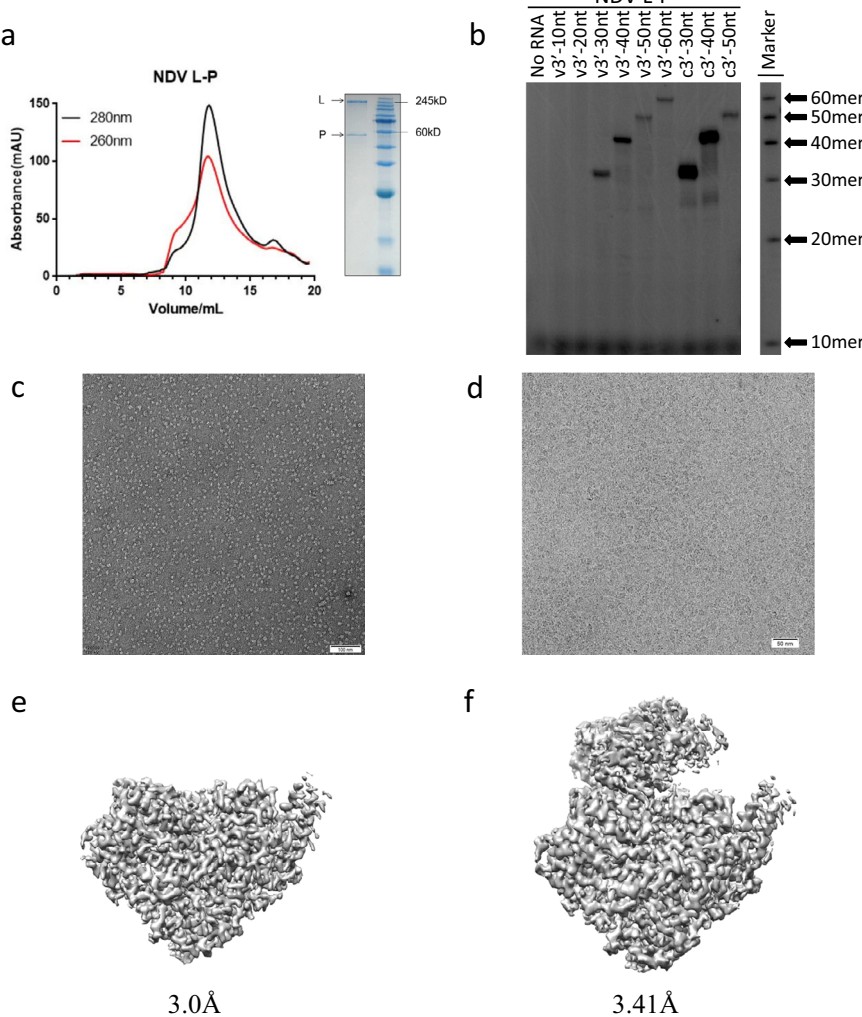

**Fig. 1 | Biochemical characterization and cryo-EM structure of NDV L-P complex. a** Elution profile of the purified NDV L-P complex on a Superose 6 Increase size-exclusion column. SDS-PAGE was performed to assess the quality of L-P complex. **b** Shown the RNA products generated from RNA synthesis reactions. **c** A representative negative staining micrograph of NDV L-P complex. **d** A representative cryo-EM micrograph of NDV L-P complex (out of 7134 micrographs). **e** The final cryo-EM density map of NDV $L_c$-P complex. **f** The final cryo-EM density map of NDV $L_f$-P complex. Source data are provided as a Source Data file. Sample preparation-related experiments including protein purification and enzymatic assays were reproduced at least three times independently.

architecture of NDV L-P complex is similar to the architecture of RNA polymerases found in other nsNS RNA viruses, where NDV harbors two additional glycine residues, both of which form GGxxG in motif B. The conserved catalytic motif C (GDN, 750–752) is located opposite to motif F. As $Mg^{2+}$ was not added during protein purification, no $Mg^{2+}$ densities were detected in our structure. Our in vitro enzymatic assay showed that L protein is activated by $Mg^{2+}$ or $Mn^{2+}$, but not by $Zn^{2+}$ (Supplementary Fig. 5a). In order to validate the amino acids within the RdRp catalytic centers, amino acids R552, I553, D641, Y645, E718, D751, N752 and GDN (750–752) were each mutated to Alanine. Our in vitro assays showed that these mutations substantially lowered the enzymatic activity except D641A and E718A (Supplementary Fig. 5b, c). Together, these results clearly demonstrated that R552, I553, Y645, D751, N752 and GDN (750–752) are critical for L protein activity.

As shown in Fig. 3c, the priming loop of NDV L adopts an RNA elongation conformation, and is directed opposite from active site similar to PIV5[5] and hMPV[28]. In contrast, the priming loop of VSV[19] L protein is positioned in the central cavity. The NDV intrusion loop is situated at the active site of the RdRp domain, which resembles PIV5, while the intrusion loop of hMPV and VSV is located away from the active site (Fig. 3d–f and Supplementary Fig. 6).

## Structural insights into the interactions between L and P proteins

To assess the oligomer structure as well as the interaction mode of the P protein, we analyzed the interface between L and P proteins. As shown in Fig. 5, the finger subdomain of L protein participates in the interaction with P proteins. The interaction between L and P is driven by both hydrophobic, hydrogen bonds and electrostatic interactions (Fig. 5a, e). The interface area of L and P is large, the total buried surface area between L and P is 6,820 $Å^2$ (Fig. 5e). The interaction can be divided into two regions: (1) on the fingers subdomain of L, four-helix bundles of P-OD use helix P1 (276–282) & P2 (271–286) fixed to L, and flexible P1 (284–299) & P2 (287–296) wrap around the surface of L to stabilize the tetrameric P on L. The β strands of P1 (284–287), P4 (308–310) and β6 (282–285) of L form a small β sheet. K284 and D287 of P1 have an electrostatic interaction with the D385, R650 and R714 of L protein (Fig. 5b). The D298 and R300 of P2 have electrostatic interactions with the residue K416, H418 and Y421 of L protein. Furthermore, I285 and L299 of P2 bind E451 and H659 of the L protein via hydrophobic interactions (Fig. 5c). (2) On the palm subdomain of the L protein, a three-helix bundle from P4 (351–399) interacts with the L protein near the putative NTP entrance channel.

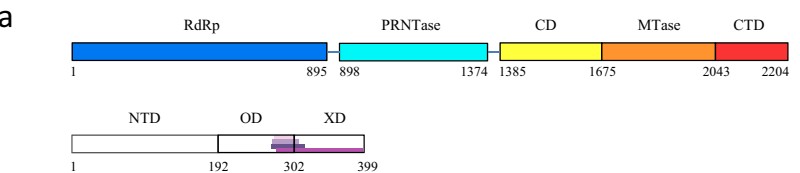

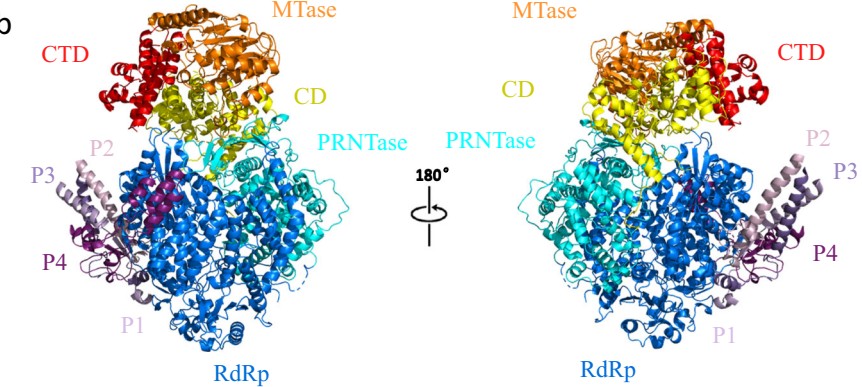

**Fig. 2 | Overall structure of NDV L-P complex. a** Schematic diagram of the domain architecture of NDV L protein divided into five parts: RdRp (blue), PRNTase (cyan), CD (yellow), MTase (orange), and CTD (red). NDV P protein is divided into three parts: NTD, OD, and XD. Each domain of NDV L is represented using a unique color and each P protein is represented using a unique color. **b** Cartoon representation of NDV L$_f$-P complex. The structures are colored by domains, and the coloring scheme is identical to that in **a**. The unresolved linker regions are connected by dashed lines. The right structure is rotated 180° along the vertical axis.

The R356, D357, R366 and K393 of P4 form electrostatic interactions with E301, D309 and H336 of L protein. In addition, L362 of P4 forms hydrophobic interactions with L299 and F306 of L protein (Fig. 5d). We did not identify any close interactions between L and P3, because the P3 monomer interacts almost exclusively with the other P monomers.

### MTase and CTD domain of L protein exhibits a unique arrangement

Next, we built an intact model of the RdRp-PRNTase-CD-MTase-CTD (Fig. 2b), and analyzed the structural features of full-length L protein of NDV with related structures. Our analysis showed that the NDV L protein contains four channels including template entrance, template exit, product exit and NTP entrance, which are essential for RNA replication/transcription (Fig. 6a, b). These channels are open in our solved structure. Through structure comparison of the four channels, we found that the PRNTase domain in the VSV[19] L protein closes the product exit (Fig. 6c), and the product exit of PIV5[5] L protein is open (Fig. 6d). Compared with the full-length structures of VSV and PIV5, the CD-MTase-CTD module of NDV adopts a unique arrangement (Fig. 6b and Supplementary Fig. 7). In contrast to VSV, the NDV CD-MTase-CTD module exhibits a small tilt relative to its RdRp domain. As a consequence, the RNA product exit channel is in open state, and continues to extend from the PRNTase domain all the way through to the CD-MTase-CTD modules on the adjacent region of thePRNTase domain. We found that the NDV MTase-CTD module is located in a position of the CD domain that is opposite to that found in PIV5. Furthermore, the MTase-CTD module is characterized by a 70° rotation with respect to its RdRp-PRNTase module. On the basis of these changes, we found that the CD domain does not stably associate with the RdRp-PRNTase module. Importantly, the movement of the CD domain causes a structural rearrangement of the MTase-CTD module. Therefore we hypothesize that the conformation of our NDV could represent an intermediate state between VSV and PIV5. This rearrangement might

be vital for the switch from initiation state to elongation state during RNA synthesis.

## Discussion

During the life cycle of the nsNS RNA virus, the RNA polymerase complex formed by L and P proteins is the vital multi-functional molecular machinery that transcribes and replicates the viral RNA genome. In our studies, we determined the atomic resolution structure of the NDV L-P complex, proved that our obtained L-P complex is functional by in vitro enzymatic assays, and validated the relation between structure and function. We found that the NDV P proteins assemble into a unique tetrameric conformation different from the hRSV, EBOV and hMPV L-P complexes, and the C-terminal adopts a unique arrangement different from previously resolved C-terminal structures of L proteins in PIV5 and VSV viruses. Our results elucidated the intermediate state in RNA elongation by comparison of NDV L-P complex with related viral L-P complex structures.

P protein is an essential polymerase cofactor that acts as a chaperone to regulate RNA synthesis. However, to date, reports of full-length P protein in complex with L proteins are limited to hRSV, EBOV, hMPV and PIV5[5,28–30]. By comparing these structures with our NDV L-P complex, we identified an NDV-specific structural re-arrangement in P proteins. In the structure of hMPV L-P complex, the P1 and P2 make extensive interacts with L in four-helix bundles region, P3 and P4 bind to the helix of P1 and P2; the C-terminal of P1 binds near the NTP entry and the C-terminal of P2 approaches the RNA template entry. Similarly, in the NDV complex structure, the four-helix bundles region from P1 and P2 extensively interact with L, while P3 and P4 bound to the helix of P1 and P2. However, in NDV structure, the C-terminal of P4 binds near the NTP entry. When we fitted the NDV P-OD tetramer model into the map of PIV5 (EMD-21095), this structure fitted well with the cryo-EM density map, indicating that P proteins structures among paramyxovirus are conserved. Thus we hypothesize that the P monomers of our NDV

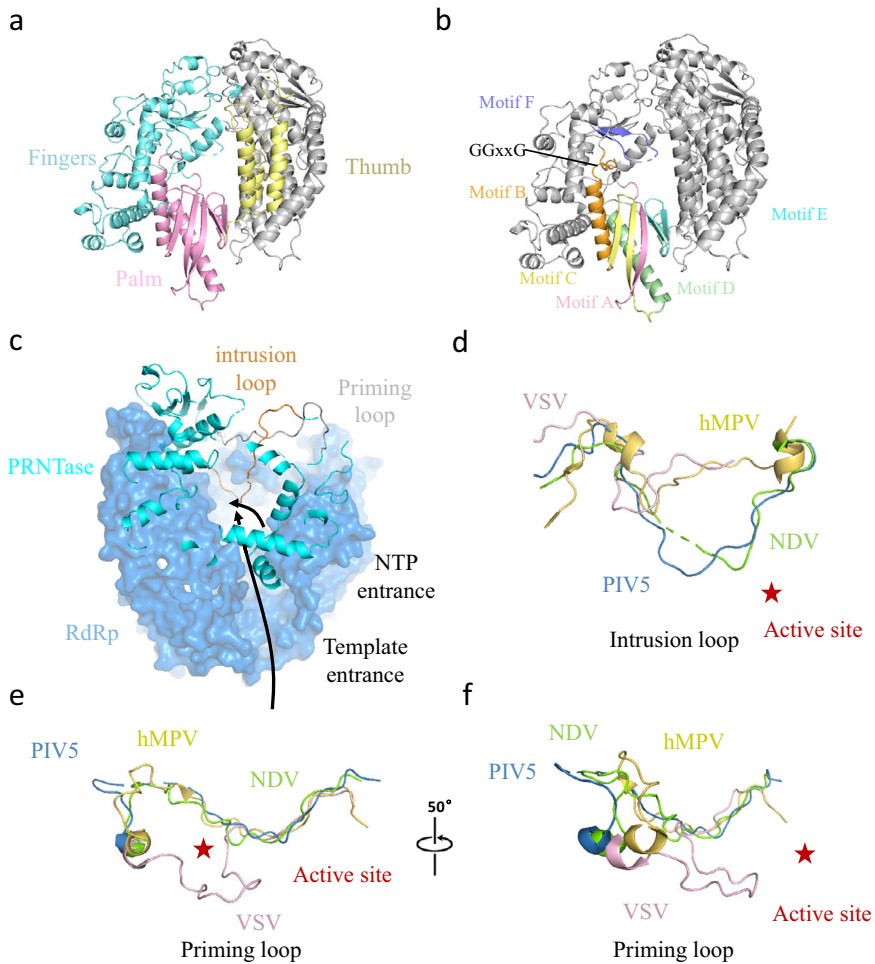

**Fig. 3 | The active site of NDV L-P complex. a** The RdRp region is highlighted in colors: the finger is shown in cyan, the thumb in pale yellow, the palm in light pink and the remaining structure of L is colored in gray. **b** Similar view as shown in **a**, and motifs A–F are highlighted in unique colors. **c** Configuration of the priming loop (gray) and intrusion loop (bright orange) in NDV L-P complex. **d** Comparison of the NDV (green), PIV5 (blue, PDB: 6V85), hMPV (yellow, PDB: 6U5O), and VSV (light pink, PDB: 5A22) intrusion loops. The catalytic site is indicated by a red star. **e, f** Comparison of the NDV (green), PIV5 (blue), hMPV (yellow), and VSV (light pink) priming loops. The catalytic site is indicated by a red star. The right region is rotated 50° along the vertical axis.

structure are present at different functional states compared to those in hMPV. During RNA synthesis, the P-OD binds to the L protein, while the C-terminal and N-terminal ends of P swing freely to perform different functions, as described in previous studies[28].

The priming loop of the L protein is involved in the formation of the first dinucleotide during de novo initiation. The intrusion loop in L protein also plays vital roles in RNA transcription, possibly by its displacement from RdRp reaction center to accommodate growing nascent RNA. When we compared the conserved motifs between NDV with other nsNS RNA viruses L protein, we found that the priming loop of NDV is similar with hMPV and PIV5, both of which retract from the active site and collapse onto the PRNTase domain, suggesting that the state of our NDV L-P complex corresponds to the RNA elongation. On the other hand, the L-P complex of VSV belongs to the pre-initiation state[19]. Variation of the distance from priming loops to the RNA polymerase reaction center (GDN) indicates that they are at distinct stages of nascent RNA elongation during transcription, namely elongation-1 for hMPV, elongation-2 for NDV and elongation-3 for PIV5 (Fig. 7 and Supplementary Fig. 4).

Based on these findings, we propose a model that captures the RNA synthesis stages of nsNS RNA virus. First, during pre-initiation of RNA synthesis (represented by VSV, Fig. 7a), the priming loop is oriented and located in the RdRp enzymatic center, while the

intrusion loop sways away from the RdRp center and towards the PRNTase domain pocket. At this stage, the nascent RNA exit channel is kept closed by the PRNTase domain, which only later will bind/cap the nascent RNA. Simultaneously, CD and CTD stack against PRNTase, resulting in MTase facing away from the nascent RNA exit channel. Once the initiation process for synthesis of the first dinucleotides is completed, L enters the elongation-1 stage (represented by hMPV, Fig. 7b). Subsequently, the movement of PRNTase opens the RNA product exit channel. The priming loop recedes to the PRNTase domain, making room for the addition of mononucleotides on the newly synthesized RNA primer. The intrusion loop moves toward the RdRp reaction center by a short distance (~3 Å, based on F1274 of hMPV). This process is followed by elongation-2 stage (represented by NDV, Fig. 7c), where the priming loop continues to move towards the PRNTase, thus providing more space for the growing nascent RNA; the intrusion loop sways toward the RdRp reaction center by a great distance (~17 Å, based on F1279 of NDV) to approach the newly synthesized RNA. Simultaneously, the CD-MTase-CTD is tilting upward in comparison to VSV. As a result, the RNA product channel continues from PRNTase to MTase, allowing the subsequent capping and methylating of the nascent RNA during this stage. Finally, at the elongation-3 stage (represented by PIV5, Fig. 7d), the priming loop fully retracts from the RdRp center and

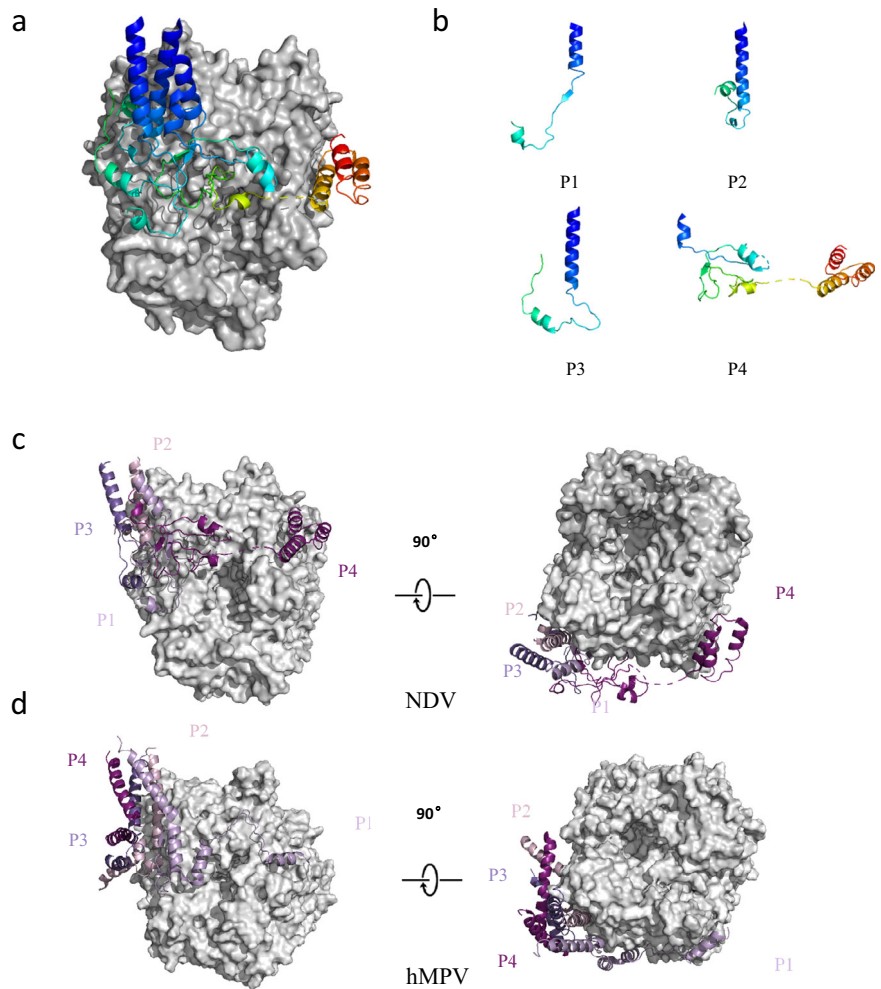

**Fig. 4 | Phosphoprotein tetramer in complex with L. a** NDV L is shown as a gray molecular surface, and P monomers are shown in cartoon. The P monomer is colored as a blue-to-red rainbow from the N- to the C-terminal end. **b** Structures of the four individual P subunits bound to L. **c**, **d** Comparison of L-P complexes between NDV and hMPV. The right region is rotated 90° along the horizontal axis.

folds back towards the cavity in PRNTase. In addition, the intrusion loop is raised (~5 Å, based on F1303 of PIV5) and moves further from the RdRp reaction center, probably facilitating to push the continuously synthesized RNA through the RNA product exit. The MTase and CTD rotate on CD domain by approx. 70° which is aided by the flexible loops connecting them, and thus orienting both MTase and CTD away from the RNA product channel. Subsequently, capped and methylated RNA is released from the RNA product exit at the end of the elongation process.

While our findings demonstrate that the L-P complex of NDV adopts a plausible intermediate state during RNA elongation, a number of questions remain. Firstly, viral RNA was not included in the complex structure, therefore it remains unknown how RNA is bound to L protein during initiation. Secondly, the structure of N terminal of the P protein remains unsolved in our final structure due to its flexible nature. Lastly, we failed to capture the complex between L-P and RNP, due to the high flexible nature of the P protein. Therefore, the precise regulatory mechanisms responsible for interaction of P protein with N protein remain to be determined. Further studies capturing the L-P-RNP complex structures should validate the hypothesized initiation and elongation processes.

In conclusion, our findings identified the structure of high-resolution NDV L-P complex, which is in a plausible RNA elongation state by structure comparison and anaylsis. Our work should

contribute to a more detailed understanding of nsNS RNA virus transcription and replication. Our insights presented here should also assist in the prevention and treatment of Paramyxoviridae and utilization of NDV in vaccine vector and cancer therapy.

## Methods

### Expression and purification of the NDV polymerase complex (L-P)

The codon-optimized sequences for NDV L protein (GenBank: ALP75897.1) and P protein (GenBank: ALP75892.1) were subcloned into pFastBac1 expression vector, and expressed in *sf*9 cells using the Bac-to-Bac expression system (Invitrogen). N-terminal 2x Strep tag followed by PreScission protease cleavage sequence and C-terminal 8x His tag was added to L protein. The double-tagged L protein and no-tagged P protein were co-expressed in *sf*9 cells by co-infection with each baculovirus at an multiplicity of infection (MOI) of ~1. After culturing for 72 h at 27 °C, cells were collected by centrifugation at 2,600 g for 20 min. Cell pellets were re-suspended in 1/10 of the original culture volume, in NDV L-P lysis buffer (50 mM Tris-HCl pH 7.8, 500 mM NaCl, 10% (v/v) glycerol, 1 mM Tris (2-carboxyethyl) phosphine (TCEP) and EDTA-free protease inhibitor cocktail) and lysed by sonication. Cell debris was pelleted by centrifugation at 18,000 g for 30 min at 4 °C and the supernatant was incubated with 2 ml Strep-Tactin XT beads for 1.5 h at 4 °C. The beads were washed three times using lysis buffer. L-P complex was eluted with a buffer containing

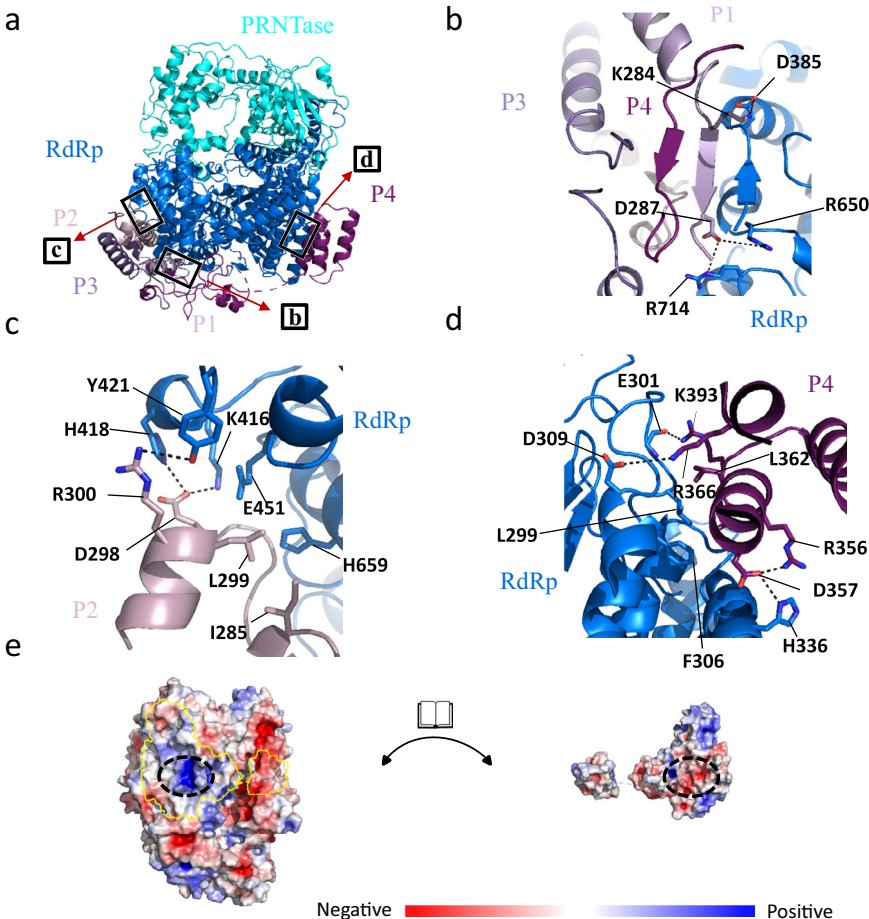

**Fig. 5 | Details of the molecular interactions between NDV L and P proteins. a** A cartoon representation of NDV L-P complex structure. The black boxes indicate the magnified view of the interactions between L and P. **b**–**d** Magnified views of interaction interfaces between L and P proteins. The representative residues involved in interaction are shown, main chain or side chains involved in hydrogen bonds or hydrophobic interactions are shown as sticks, with oxygen and nitrogen atoms shown in red and blue, respectively. Electrostatic interactions are depicted as dashed lines. **e** L and P proteins are shown as molecular surfaces colored by electrostatic potential from red to blue, negative to positive, respectively. The surface of each protein is shown in an open-book representation. The approximate binding interface is outlined in yellow. A region of electrostatic complementarity is highlighted with a dashed oval.

50 mM Tris-HCl pH 7.8, 500 mM NaCl, 10% (v/v) glycerol, 1 mM TCEP and 50 mM biotin. To further purify the complex, the eluted sample was subjected to Superose 6 Increase 10/300 GL (GE Healthcare) with the same elution buffer without biotin or glycerol. The purified protein complex was flash-frozen in liquid nitrogen and stored at −80 °C until use.

## Cryo-EM sample preparation and data collection

To prepare cryo-EM samples, 4 µl of purified NDV L-P complex in a buffer containing 50 mM Tris-HCl pH 7.8, 500 mM NaCl, 1 mM TCEP was applied to each grid. Amorphous alloy films (R1.2/1.3, Au, 300 mesh) were glow-discharged for 50 seconds prior to the application of the complex at 1 mg/ml concentration. The grids were blotted for 3 s at ~100% humidity and plunged into liquid ethane using an FEI Vitrobot Mark IV. The samples were loaded onto the FEI Titan Krios transmission electron microscope at 300 kV with a GIF-Quantum energy filter (Gatan) and a Gatan K2-summit detector for data collection. Automatic data collection used SerialEM software (http://bio3d.colorado.edu/SerialEM/). The nominal magnification of ×165,000 corresponds to a calibrated pixel size of 1.04 Å at the specimen and a dose rate of 10 e⁻/pixel/s. Each image was exposed for 6.5 s to obtain an accumulative dose of ~60 e⁻/Å², fractioned into 32 frames. The final defocus range of all images sets was approximately −1.5 to −2.5 µm. A total of 7134 micrographs were collected.

## Image processing

Beam-induced motion and anisotropic magnification were corrected by the program Motioncorr2 with a 5 × 5 patch. Initial contrast transfer function (CTF) parameters for each micrograph were estimated with Gctf[36]. Micrographs with resolution better than 4 Å were selected for subsequent data processing. 50 micrographs were selected for manual particle picking and ~10k particles from manual picking were subjected to 2D classification to generate templates for auto-picking against all the micrographs. Based on the template ~4,238k particles were selected for reconstruction.

The program RELION3.0.8[37] was used to process and reconstruct the structure. After 2D classification, a subset of ~2,589k particles was selected and further subjected to 3D classification with the model from 3D initial model in RELION[37]. The selected particles were divided into five classes through 3D classification. The best class containing 455k particles was selected for 3D refinement and we obtained a density map with a resolution of 3.0 Å.

Next, to separate the full-length particles and obtained an initial full-length model, we used focus 3D classification. Using the initial full-length model as template, particles were divided into five classes through 3D classification. One of the classes was characterized by the full-length particle, from which we obtained a 3.41 Å map after refinement. Using similar procedures, we determined the 3.25 Å structure of $L_c$-$P_e$.

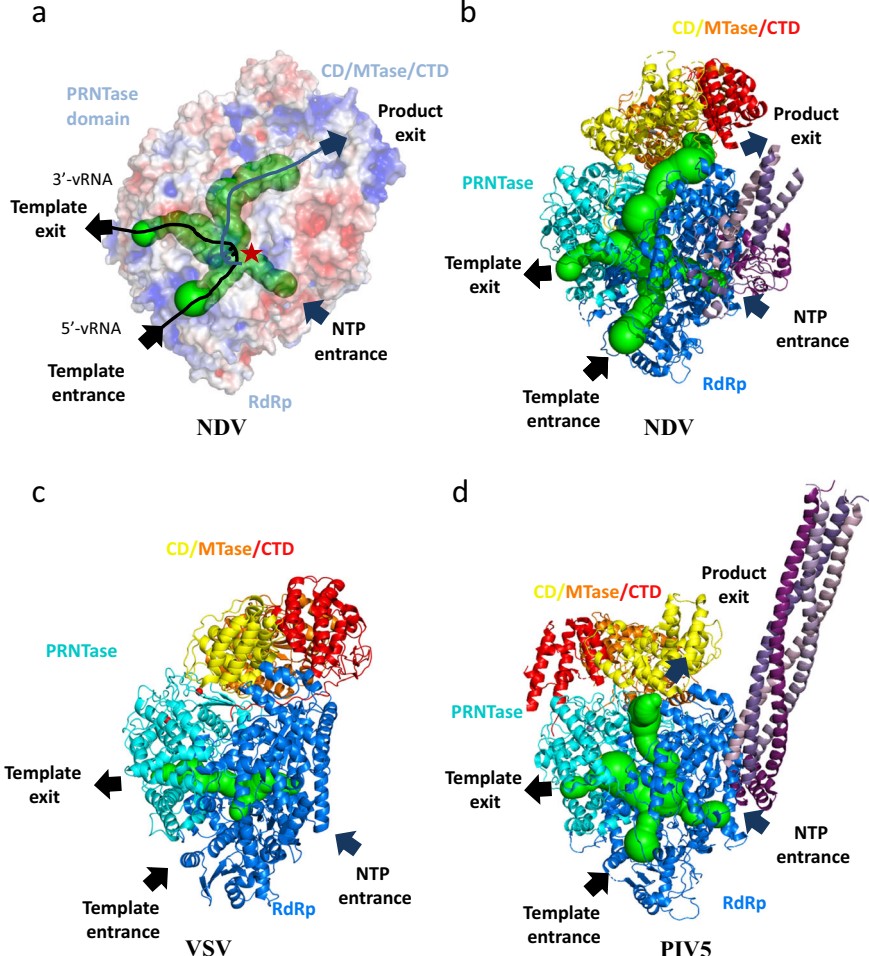

**Fig. 6 | Structural comparison of NDV, VSV and PIV5 L-P complexes. a** Tunnels within the NDV L-P structure for RNA synthesis. The template entrance and exit, the NTP entrance and product exit are shown. **b**–**d** Comparison of the NDV **b**,VSV **c**, and PIV5 **d** were aligned based on the RdRp domain (blue) and PRNTase domain (cyan). The CD (yellow), MTase (orange) and CTD (red) have large rearrangement between the three structures.

## Model building and refinement

To build and refine our models, the two reconstructed maps of the $L_f$-P complex and the $L_c$-P complex at 3.41 Å and 3.0 Å respectively, were used for model building and refinement. The program AlphaFold[38] was used to predict the RdRp and PRNTase domain of L protein. Program UCSF chimera[39] was used to position the coordinates of the model into the 3.0 Å map. Further optimization of the positioning of the side chains were performed using the program COOT[40] and the real-space refinement was performed using the program PHENIX[35]. We used the full-length P model predicted by AlphaFold[38] and positioned it to the cryo-EM density map by UCSF chimera[39] followed by optimization in COOT[40], with the structure of PIV5 L-P complex (PDB:6V85) for guidance. PHENIX[35] was applied to refine the structure of NDV $L_f$-P complex. Based on the $L_c$-P complex, the $L_f$-P complex was built based on the predicted model of CD/MTase/CTD domain by AlphaFold[38], followed by manual optimization in COOT[40] and real-space optimization in PHENIX[35]. The structure of $L_c$-$P_e$ was determined and refined using the same method as was used for $L_c$-P.

## In vitro enzymatic assay of the NDV L-P complex

RNA oligonucleotides (see Supplementary Table 3 for detailed sequence information) were chemically synthesized and purified (Ruibiotech, China) as the templates. NTPs were purchased from Takara Biomedical Technology, China. Radioactive isotope-labeled nucleotides [α-32P]-ATP and [γ-32P]-ATP were purchased from PerkinElmer, USA. The reaction mixtures contained 2.4 μM RNA template,

1.5 μM NDV L-P complex, NTPs (124 μM UTP, 124 μM GTP, 887 μM CTP, 33.36 μM ATP) and 165 nM of [α-32P]-ATP (3000 Ci/mmol), and reaction buffer (20 mM Tris pH 8.0, 50 mM NaCl, 2 mM DTT, 0.5% (v/v) Triton X-100, and 6 mM MgCl$_2$) in a final volume of 10 μl. Before added to the reaction mixture, the template was denatured at 75 °C for 5 min before being slowly cooled to 4 °C in a PCR machine. The reaction mixtures were incubated at 30 °C for 3 h, then stopped by addition of 10 μl RNA Gel Loading Dye (2×) (Thermo Scientific) containing 47.5% (v/v) formamide as denaturing agent. The RNA products were resolved on denaturing 7 M urea 20% polyacrylamide gel electrophoresis in 0.5× TBE buffer, and analyzed by autoradiography through phosphorimaging (Typhoon FLA 7000, GE Healthcare Bio-Sciences). The lengths of the RNA products were determined by comparing with a mixture of various lengths of synthesized marker RNA labeled with T4 polynucleotide kinase (PNK) (New England Biolabs) and [γ-32P]-ATP (3000 Ci/mmol). The sequences for the ladders are the same as Supplementary Table 3 vRNA. To characterize the effect of metal ions on enzymatic activity, MgCl$_2$, MnCl$_2$ and ZnCl$_2$ were each added to the reaction mixture at a final concentration of 5 mM. The template for mutant experiments is 3'-40nt cRNA. To confirm obtained results, all assays were performed independently at least twice.

**Figure preparation.** All the representing model and cryo-EM density maps were generated using programs UCSF Chimera[39], and PyMOL[41]. Programs Clustal X[42] and ESPript[43] were used to align multiple sequences.

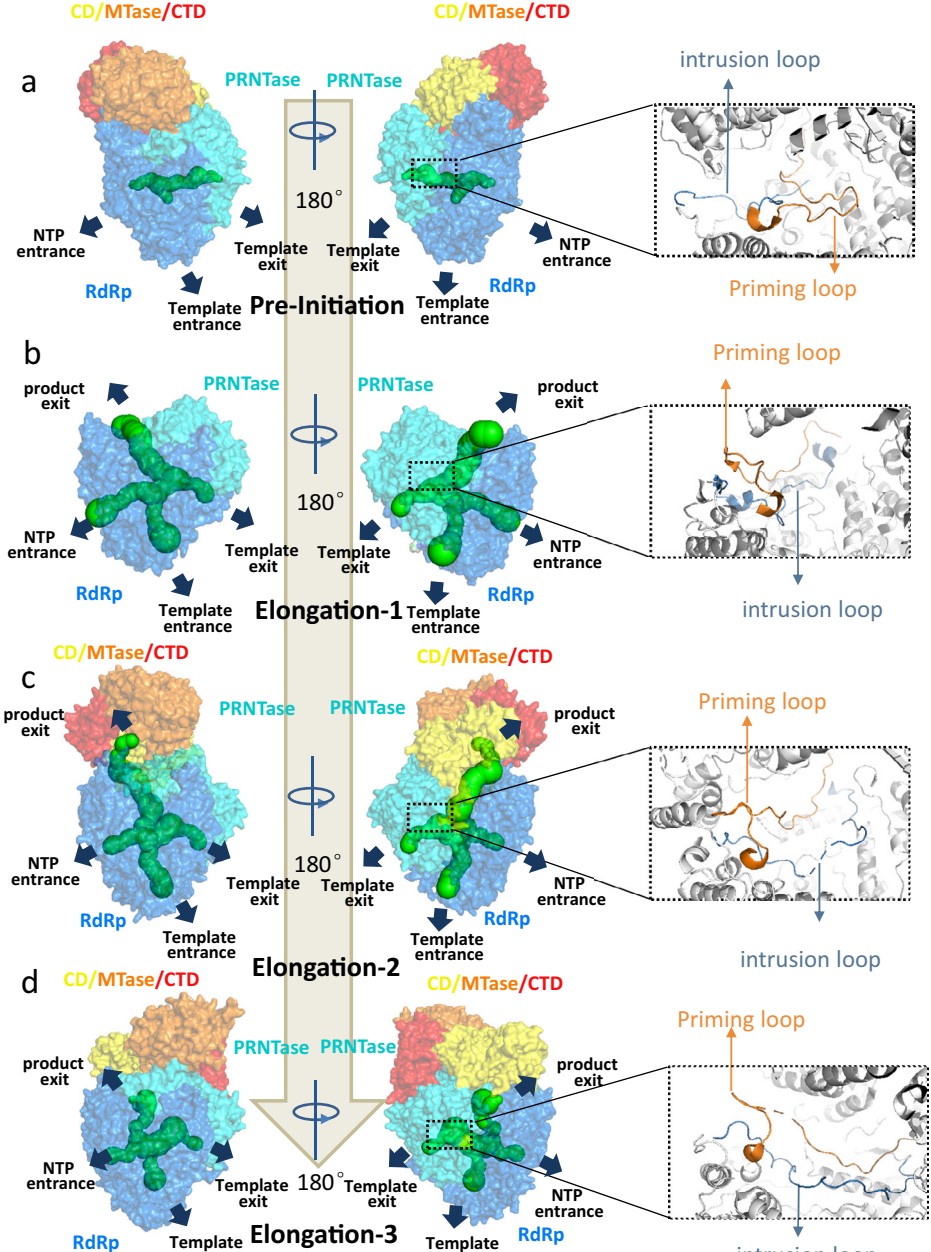

**Fig. 7 | Model of initiation and elongation for L-P complex in RNA synthesis. a** In pre-initiation (represented by VSV), the priming loop is oriented in active site while the intrusion loop is towards the PRNTase domain. The product exit channel is blocked by PRNTase domain. The CD-MTase-CTD module is near the product exit channel. **b** In elongation-1 stage (represented by hMPV missing CD-MTase-CTD module), the priming loop is retracted from the active site and the intrusion loop is also located in the PRNTase domain. The PRNTase domain moves to expose RNA product exit channel. **c** In elongation-2 stage (represented by NDV), The priming loop is retracted from the active site and the intrusion loop extends into the active site. The CD-MTase-CTD module tilts upward, and the product channel is continuous through PRNTase and MTase. **d** In elongation-3 stage (represented by PIV5), the priming loop is retracted further from the active site; the intrusion loop raises upward from the active site. The MTase-CTD modules was rotated by -70 degree, orienting MTase and CTD away from the product channel.

## Reporting summary

Further information on research design is available in the Nature Portfolio Reporting Summary linked to this article.

## Data availability

The structure of NDV L-P complex has been deposited at the Protein Data Bank (PDB), (accession codes $L_f$-P: "7YOU", $L_c$-P: "7YOT", $L_c$-$P_e$: "7YOV"). The cryo-EM density map of NDV L-P complex has been deposited at the Electron Microscopy Data Bank (accession codes $L_f$-P: "EMD-33987", $L_c$-P: "EMD-33986", $L_c$-$P_e$: "EMD-33988"). Source data are provided with this paper. The coordinates we used for structural analysis includes PDB: "6V85", "6U5O", "5A22". Source data are provided with this paper.

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

## Acknowledgements

We thank Boling Zhu, Lihong Chen and Xiaojun Huang for cryo-EM data collection, the Center for Biological imaging (CBI) in Institute of Biophysics (IBP) for conducting all EM work. We thank Hongjie Zhang for the guidance in handling radiolabeled chemicals. We thank Bei Yang for assistance with our cell biology experiments. We thank Yan Wu for his research assistant service. We thank Torsten Juelich for his contribution in manuscript revision. We thank Xianjin Ou for technical assistance during the fermentation/protein preparation steps. All research described in this article is supported by the National Basic Research Program (Grant No. 2017YFC0840300) to Y.C.; it is also supported by National Key Research and Development Program (Grant No. 2020YFA0707503), the Strategic Priority Research Program of the Chinese Academy of Sciences (XDB37030200), National Basic Research Program (Grant No. 2016DDJ1ZZ17) to X.L.

## Author contributions

J.C. and Y.C. designed the study. X.F., J.C., and H.K. performed protein purification and cryo-EM sample preparation. J.C., X.F., and C.W. performed the cryo-EM data collection. X.F., Y.C., and J.C. conducted biochemical experiments and mutant experiments; J.C., W.F., L.W., and Y.C. determined and refined the structure. J.C. and X.F. wrote the manuscript together with X.L. and Y.C. Z.R. supervised the project.

## Competing interests

The authors declare no competing interests.
