## [Peer Review File · Nature Communications]

Structure of the Newcastle Disease Virus L protein in complex with tetrameric phosphoproteinREVIEWER COMMENTS

Reviewer #1 (Remarks to the Author):

In this manuscript Cong and colleagues present the cryo-EM structure of the Newcastle Disease Virus (NDV) RNA polymerase (L) in complex with phosphoprotein (P). The authors succeed in resolving the complete structure of L, including the RNA-dependent RNA polymerase (RdRp) domain, the polyribonucleotidyl transferase domain (PRNTase), connector domain (CD), methyltransferase domain (MT) and C-terminal domain (CTD). They observe four copies of P which are only partially resolved and show different conformations. They describe in detail the interactions between NDV L and P proteins and compare their L-P structure with that of other negative sense RNA viruses such as hMPV, VSV and PIV5. The authors conclude by proposing a model for the initiation and elongation for an L-P complex in RNA synthesis.

This study is another in the line of several recent studies reporting structures of non-segmented negative sense RNA viruses. It highlights some interesting differences between NDV L-P and these previous structures but overall it does not substantially improve our understanding of the mechanisms used by these RNA polymerases to transcribe viral genes and replicate the viral genome. Nevertheless, as the study reports the first NDV L-P structure it will be an important addition to the literature. The data appear solid and support the conclusions but the presentation could be improved (see specific points).

Specific points:

1. The authors should tone down the paper in terms of referring to their structure as an elongation state (see, for example, title, lines 300-301, 310-312). While the proposal that the observed conformation could approximate an elongation intermediate state is plausible the observed structure contains no RNA template and no RNA product. Hence it is unlikely to represent a true elongation state polymerase.
2. Fig. 3c. There are no detectable products from the 10nt or 20nt templates; is there any explanation for this? There is also no information given on how these products have been characterised to ensure they are authentic transcription products. The experiment does not include controls such as L active site mutant or activity in the absence of P. In addition, should "c5'-60nt" read as "c5'-50nt" at the top of the gel (last lane); the observed band appears to migrate close to the 50mer marker.
3. Supplementary Fig. 3. Specify the length of template RNA used in the assay. I would assume it was either the v5'-40nt or v5'-50nt in which case what is the reason for the product migrating between the 40mer and 50mer markers? In addition, in panel b the sizes of products are clearly different, some lanes containing multiple bands; can the authors comment on this? There is no information on the quality of the mutant L-P complexes used – ideally protein gels should be shown.

Minor points:

4. The figures are called in a confusing way; please co-ordinate figures/figure panels to match the order being called in the text.
5. Fig1a. Please check the molecular weight marker annotation; the L protein is previously stated to be 250 kDa but in the gel it is close to the 175 kDa marker.
6. Fig. 3. In the legends, specify PDB ID for the VSV and PIV5 structures used for comparison.
7. Fig. 4c and d. Check that the arrows indicating 90-degree rotation are correct.
8. Fig. 5. In panel a the letters in boxes obscure the structural detail underneath; could the lettering be moved to the side of the figure? In panels b-d here are several residues that are hard to see – please re-label them. In panel e the right part should be moved up to match the location on the left molecule.
9. Fig. 6d. Top of the panel is cut off.
10. Fig. 7. This figure needs editing to avoid clashes of labels.
11. Supplementary Fig. 1. The resolution range colouring appears off; perhaps the mask or resolution range is too tight. From the colouring the RdRp domain appears of the lowest resolution; is this correct?
12. Supplementary Table 3. Specify the 5' and 3' ends of RNAs to avoid confusion.
13. Line 395. GTP is mentioned twice; I assume one should be CTP.
14. The authors might wish to include the Ebola L protein (PMID: 36171293) for comparison. This reviewer had no access to the structural data and therefore the quality of the maps/models has not been assessed.

Reviewer #2 (Remarks to the Author):

The manuscript from Cong et al reports the cryo-EM structure of the NDV polymerase complex, which consists of the L protein and tetrameric P protein. The complex is demonstrated to be active via an RNA synthesis assay, and the cryo-EM studies are performed well with reasonably good statistics for the maps and models. Overall, the structure of the NDV LP complex is most similar to the previously determined PIV5 LP complex, which is expected given that both NDV and PIV5 are paramyxoviruses. There are some differences between the PIV5 and NDV LP complexes, namely the arrangement and position of the P protomers and the CD/MTase/CTD domain. Because of these differences, the authors claim that their structure represents a previously uncharacterized intermediate of the elongation state (as claimed in the title). However, it seems just as likely that the structural differences may be due to differences in the sequences of LP proteins in NDV and PIV5. If the latter is true, then the structural studies, and manuscript, do not have sufficient novelty for publication in Nature Communications. To be considered for publication in Nature Communications, the authors need to provide convincing evidence that their structure does in fact represent a novel elongation intermediate state.

Minor comment:

-In several instances the authors refer to the cryo-EM map as electron density. However, unlike X-ray crystallography, which does produce electron density maps, cryo-EM produces electrostatic potential maps.

Reviewer #3 (Remarks to the Author):

This paper reported the NDV L/P complex in three states, demonstrating the interaction between NDV L and P proteins. The structural comparison to hMPV, VSV, and PIV5 polymerases also provides insights into the RNA synthesis mechanism in NNS RNA viruses. Overall, it is a paper with solid experimental data.

Major comments:

Line 24-27: the interaction between P and L can not indicate the elongation state of the complex (priming loop and the overall conformation.)

This paper talked about the priming loop and the intrusion loop: the priming loop is related to the initiation of RNA synthesis; however, the function of the intrusion loop is not clearly illustrated. The HR motif is located on the intrusion loop, which catalyzes the capping of the mRNA. Any evidence indicates the relationship between intrusion loop and initiation, as mentioned in lines 260-262?

Supplementary Figure 7: The sequence alignment of NDV L/P with PIV5, hMPV, and VSV. In this Figure, the labels of K-D-K-E is not clearly highlighted. Besides, the motifs of GXXT and K-D-K-E are not well aligned, especially with VSV L. Furthermore, P shares less similarity, making the alignment in Supplementary Figure 7b meaningless. In this case, the NTD, OD, and XD of P should be aligned separately.

Figures 6 and 7: the NDV L/P complex is similar to VSV L/P complex in which the CTDs are on the opposite side of CD-MTase compared to that of PIV5 L/P complex. In Figure 6, the CD and MTase domains locate similarly at the top of the RdRp and PRNTase domains, while the CTD changes a lot between NDV/VSV and PIV5. In this paper, the author also claimed that the NDV L/P represents an intermediate state between VSV/hMPV and PIV5 L/P (line 228). However, there is no evidence showing that the CTD will rotate to the other side of CD-MTase during RNA synthesis in the same polymerase. Besides, the function of the intrusion loop conformation changes is unclear.

Minor comments:

The mapped part of the P protein in Figure 2a and Supplementary Figure 2a should be different. Figure 3c, should they be 3' instead of 5'. And is the last one c3'-50 nts instead of 60 nts as the table 3.

Supplementary Figure 3: which RNA template was used in those figures? 40 nts? The bands were between 40 mer and 50 mer.

Line 13: "Newcastle disease virus (NDV)" NDV abbreviation

Line 43: 15.2k nts

Line 103: the concentration of the sample is 1mg/ml, while in line 342 it says 0.8 mg/ml. Which one is correct?

Line 192-205: Please specify Figures 5b, 5c, and 5d in corresponding locations, not only the last one. Figure 5e is not mentioned in the text.

Line 280: PRNTase moves away??

Line 387: Lf-Pe should be Lc-Pe

Line 395: NTPs (... , 124 nM GTP, 887 nM GTP???) AND 165 nM of [α -³²P]-ATP (3000 Ci/mmol).

Two GTPs? And is the ATP 10 mCi/ml? Is any cold ATP added to the reaction?

Line 405-407: what are the sequences for the ladders? Are they the same as the products?

Reviewer #1 (Remarks to the Author):

In this manuscript Cong and colleagues present the cryo-EM structure of the Newcastle Disease Virus (NDV) RNA polymerase (L) in complex with phosphoprotein (P). The authors succeed in resolving the complete structure of L, including the RNA-dependent RNA polymerase (RdRp) domain, the polyribonucleotidyl transferase domain (PRNTase), connector domain (CD), methyltransferase domain (MT) and C-terminal domain (CTD). They observe four copies of P which are only partially resolved and show different conformations. They describe in detail the interactions between NDV L and P proteins and compare their L-P structure with that of other negative sense RNA viruses such as hMPV, VSV and PIV5. The authors conclude by proposing a model for the initiation and elongation for an L-P complex in RNA synthesis.

This study is another in the line of several recent studies reporting structures of non-segmented negative sense RNA viruses. It highlights some interesting differences between NDV L-P and these previous structures but overall it does not substantially improve our understanding of the mechanisms used by these RNA polymerases to transcribe viral genes and replicate the viral genome. Nevertheless, as the study reports the first NDV L-P structure it will be an important addition to the literature. The data appear solid and support the conclusions but the presentation could be improved (see specific points).

Our response: We thank the reviewer for the summary and for the constructive suggestions on how to improve our manuscript. We have now revised our manuscript in line with the reviewers' suggestions.

Specific points:

1. The authors should tone down the paper in terms of referring to their structure as an elongation state (see, for example, title, lines 300-301, 310-312). While the proposal that the observed conformation could approximate an elongation intermediate state is plausible the observed structure contains no RNA template and no RNA product. Hence it is unlikely to represent a true elongation state polymerase.

Our response: We thank the reviewer for this comment. To date, no structures of nsNSV L proteins bound to template RNA are available, with several conformations of apo L-P complexes reported, suggesting that conformational rearrangements take place when the polymerase transits from the pre-initiation to the elongation state (Nat Rev Microbiol. 2021 Mar;19(3):220.). Taking the insights from that review into account, we strongly believe that the priming/intrusion loop and the overall conformation in our structure are a strong indication that the NDV L-P complex constitutes an elongation intermediate.

While we feel that our model is not overly speculative, we understand that without the RNA complex, it remains just that; however, the function of a model is to provide a stepping stone for future inquiries. To address the concern of the reviewer, we have changed the title of our paper to "Structure of the Newcastle Disease Virus L protein

in complex with tetrameric phosphoprotein". In addition, we have now rephrased the presentation at line 300-301 and 310-312 in our revised manuscript.

2. Fig. 3c. There are no detectable products from the 10nt or 20nt templates; is there any explanation for this? There is also no information given on how these products have been characterised to ensure they are authentic transcription products. The experiment does not include controls such as L active site mutant or activity in the absence of P. In addition, should "c5'-60nt" read as "c5'-50nt" at the top of the gel (last lane); the observed band appears to migrate close to the 50mer marker.

Our response: We thank the reviewer for this comment, and agree that these controls are necessary. We have now performed an EMSA assay to screen for binding templates prior to the *in vitro* enzymatic assay. We found that only RNA with a minimum of 30nt stably binds the L-P complex. On the basis of this result, we believe that low binding affinity is the reason for the absence of nascent RNA products with 10 and 20nt RNA as template.

Based on the fact that the RNA product is of the same length as the corresponding template, we are inclined to conclude that the product in our assay is the RNA replication product.

In our experiment, we attempted to obtain native L protein in the absence of P protein. However, when expressed alone, the L protein appears to be both unstable and insoluble. As a consequence, applying L protein alone was not feasible in our *in vitro* assay.

We thank the reviewer for pointing out this error, we have now corrected Fig. 3c to "c5'-50nt" in the revised version of our manuscript.

3. Supplementary Fig. 3. Specify the length of template RNA used in the assay. I would assume it was either the v5'-40nt or v5'-50nt in which case what is the reason for the product migrating between the 40mer and 50mer markers? In addition, in panel b the sizes of products are clearly different, some lanes containing multiple bands; can the authors comment on this? There is no information on the quality of the mutant L-P complexes used – ideally protein gels should be shown.

Our response: The template RNA used in this assay is c3'-40nt RNA, due to the fact that its product is clearer to observe. The marker we used was 10/20/30/40/50/60 bp

3'-vRNA. Two reasons might explain why the 40nt RNA product was located between 40-50mer markers: 1) The actual molecular weight for the RNA product from the v40 template was 13.21 kDa, while the molecular weight for 40mer and 50mer markers are 12.85 kDa and 16.02 kDa, respectively; 2) the difference of loading volume for RNA product (10 μ L) and RNA marker (1 μ L).

Currently we are not completely clear why some lanes contain multiple bands, although our replicated assay has confirmed this phenomenon. However, previous findings indicated that the L-P complex is capable of modifying the 3' terminus of the template in addition to engaging in *de novo* initiation of RNA synthesis (Nature. 2020 Jan;577(7789):275-279), which might help explain this similar phenomenon in our assay.

To confirm the quality of the mutant L-P complexes, we have included the SDS-PAGE for the mutants in supplementary Figure 5c. The purity of L-P mutants was comparable to that of native L-P complex.

Minor points:

4. The figures are called in a confusing way; please co-ordinate figures/figure panels to match the order being called in the text.

Our response: We thank the reviewer for this suggestion. We have now adjusted the order of the figures/supplementary figures and matched them with the text in our revised manuscript.

5. Fig1a. Please check the molecular weight marker annotation; the L protein is previously stated to be 250 kDa but in the gel it is close to the 175 kDa marker.

Our response: The reason that the L protein (~250 kDa) is close to the 175 kDa marker in the gel is because the top band next to L protein is the largest band. In our revised manuscript, we have redone the SDS-PAGE with a new protein marker that includes a 245 kDa band, thus reflecting the molecular size of our L protein better. (Fig. 1a)

6. Fig. 3. In the legends, specify PDB ID for the VSV and PIV5 structures used for comparison.

Our response: We have now added the PDB ID of VSV, PIV5 and hMPV in the figure legends.

7. Fig. 4c and d. Check that the arrows indicating 90-degree rotation are correct.

Our response: We thank the reviewer for pointing this out. We have corrected the direction of the arrow in the figure.

8. Fig. 5. In panel a the letters in boxes obscure the structural detail underneath; could the lettering be moved to the side of the figure? In panels b-d here are several residues that are hard to see – please re-label them. In panel e the right part should be moved up to match the location on the left molecule.

Our response: We thank the reviewer for the suggestion to make the figures clearer. We have moved the lettering, re-labeled the residues in panels b-d and moved right part in panel e to match the location on the left.

9. Fig. 6d. Top of the panel is cut off.

Our response: We thank the reviewer for pointing this out. We have now redrawn Fig.

6d.

10. Fig. 7. This figure needs editing to avoid clashes of labels.

Our response: We have re-edited the labels to avoid the clashes.

11. Supplementary Fig. 1. The resolution range colouring appears off; perhaps the mask or resolution range is too tight. From the colouring the RdRp domain appears of the lowest resolution; is this correct?

Our response: We thank the reviewer for pointing this out. We have now checked that the solvent mask we used was a local mask. Therefore, we have now used the global mask as solvent mask to run local resolution again, and replaced the former figure.

12. Supplementary Table 3. Specify the 5' and 3' ends of RNAs to avoid confusion.

Our response: We thank the reviewer for pointing this out. We have now added the 5' and 3' labels to each sequence to specify the RNA ends more clearly, hopefully avoiding confusion for our readers.

13. Line 395. GTP is mentioned twice; I assume one should be CTP.

Our response: We thank the reviewer for pointing this out. The second GTP was actually CTP and we have corrected this issue in the methods part describing the *In vitro* enzymatic assay.

14. The authors might wish to include the Ebola L protein (PMID: 36171293) for comparison.

This reviewer had no access to the structural data and therefore the quality of the maps/models has not been assessed.

Our response: As the suggestion of reviewer, we have compared the L-P structures of Ebola, hMPV and NDV, and found that the key loops (priming and intrusion loop) and the conformation of P tetramers were similar (except that Ebola has a more intact P-OD domain). Therefore, we did not add new figures for specific comparison of NDV and Ebola. Instead, we have now included this information in the introduction/result/discussion sections.

P.S. We have sent the maps/models to the editor while this manuscript was being reviewed.

Reviewer #2 (Remarks to the Author):

The manuscript from Cong et al reports the cryo-EM structure of the NDV polymerase complex, which consists of the L protein and tetrameric P protein. The complex is demonstrated to be active via an RNA synthesis assay, and the cryo-EM studies are performed well with reasonably good statistics for the maps and models. Overall, the structure of the NDV LP complex is most similar to the previously determined PIV5 LP complex, which is expected given that both NDV and PIV5 are paramyxoviruses. There are some differences between the PIV5 and NDV LP complexes, namely the arrangement and position of the P protomers and the CD/MTase/CTD domain. Because of these differences, the authors claim that their structure represents a previously uncharacterized intermediate of the elongation state (as claimed in the title). However, it seems just as likely that the structural differences

may be due to differences in the sequences of LP proteins in NDV and PIV5. If the latter is true, then the structural studies, and manuscript, do not have sufficient novelty for publication in Nature Communications. To be considered for publication in Nature Communications, the authors need to provide convincing evidence that their structure does in fact represent a novel elongation intermediate state.

Our response: We thank the reviewer for the constructive comments. We agree that the structures of NDV and PIV5 are similar, making PIV5 the important comparison with NDV. We have now adjusted our expressions regarding this conformational change. However, we do not fully agree with the reviewer. As mentioned in a previous review (Nat Rev Microbiol. 2021 Mar;19(3):220.), several captured conformations of apo L-P complexes suggest that conformational rearrangements take place when the polymerase transits from the pre-initiation to the elongation state. To test this, we combined the conformations of NDV and other typical L-P structures for comparison, and proposed a new model that is based on these findings. Our proposed model not only includes the rearrangement of CD/MTase/CTD domain, but also the dynamics of internal channels within L proteins, flipping of key loops (the priming loop and intrusion loop), and the changes of P protein conformation. Therefore, we are confident that our model constitutes a fairly accurate description of reality. However, we understand that the claim that it presents an “intermediate state” is possibly on the speculative side, since no RNA complex was obtained in our findings. We have revised these expressions in our manuscript, and are now using the “arrangement” suggested by reviewer to replace the “conformation”. We also rephrased our title as “Structure of the Newcastle Disease Virus L protein in complex with tetrameric phosphoprotein”. Hopefully this will make our presentation more objective in the eyes of the readers.

The reviewer mentioned “it seems just as likely that the structural differences may be due to differences in the sequences of LP proteins in NDV and PIV5”. We beg to differ on this point. The structure of NDV MTase-CTD module is characterized by a 70° rotation relative to the RdRp-PRNTase module, when compared with PIV5. We believe that this deflection is caused by its natural multiple functional states rather than by mere amino acid sequence variance. The CD-MTase-CTD module of nsNSV L protein is flexible and therefore missing in many L-P complex structures. As we observed, the NDV L_f-P particles occupy only 1.8% of the total particles, so other conformations of full-length L protein were not captured due to its flexibility. Therefore, we solved the NDV full-length L-P structure in a single conformation. We believe the particles in this conformation are more stable than the rest particles.

In respect to the novelty of our findings, we have resolved the high-resolution structure of the first NDV full-length L protein and P protein tetramer complex, and elucidated in detail the interaction interface between L and P, which have not been achieved simultaneously before. Our model is based on NDV and other nsNSV structures, and thus constitutes a major step towards a more detailed understanding of nsNSV RNA synthesis.

Minor comment:

-In several instances the authors refer to the cryo-EM map as electron density. However, unlike X-ray crystallography, which does produce electron density maps, cryo-EM produces electrostatic potential maps.

Our response: We thank the reviewer for pointing this suggestion. We have rephrased “electron density” to “cryo-EM density map” in our revised manuscript.

Reviewer #3 (Remarks to the Author):

This paper reported the NDV L/P complex in three states, demonstrating the interaction between NDV L and P proteins. The structural comparison to hMPV, VSV, and PIV5 polymerases also provides insights into the RNA synthesis mechanism in NNS RNA viruses. Overall, it is a paper with solid experimental data.

Our response: We thank the reviewer for summarizing our main findings and positive comments.

Major comments:

Line 24-27: the interaction between P and L can not indicate the elongation state of the complex (priming loop and the overall conformation.)

Our response: We thank the reviewer for the constructive suggestion, and agree that the L-P interaction mode alone is not sufficient to lead to the “elongation state” statement. We have modified the phrases, so that the C-terminal module arrangement, priming/intrusion loop conformation and P protein interaction are all involved to lead to this conclusion.

This paper talked about the priming loop and the intrusion loop: the priming loop is related to the initiation of RNA synthesis; however, the function of the intrusion loop is not clearly illustrated. The HR motif is located on the intrusion loop, which catalyzes the capping of the mRNA. Any evidence indicates the relationship between intrusion loop and initiation, as mentioned in lines 260-262?

Our response: We thank the reviewer for bringing up the role/importance of the intrusion loop. The introduction and description for the function of intrusion loop was only reported in the cryo-EM structure of PIV5 L-P complex (Proc Natl Acad Sci U S A. 2020 Mar 3;117(9):4931-4941.). “Displacement of the intrusion loop is required to accommodate RNA in the active site, suggesting a possible tug-of-war between the priming loop and intrusion loop that could regulate transcription initiation.”. We therefore combined the result of PIV5 to propose our model. We have now rephrased our sentences at the former lines 260-262 in our revised manuscript.

Supplementary Figure 7: The sequence alignment of NDV L/P with PIV5, hMPV, and VSV. In this Figure, the labels of K-D-K-E is not clearly highlighted. Besides, the motifs of GXXT and K-D-K-E are not well aligned, especially with VSV L. Furthermore, P shares less similarity, making the alignment in Supplementary Figure 7b meaningless. In this case, the NTD, OD, and XD of P should be aligned separately.

Our response: We thank the reviewer for the constructive suggestion and pointing out the misalignment in the figure. We have modified the Figure 7 (now as Supplementary Figure 2 in revised manuscript) as suggested, highlighting the labels

for L proteins. We attempted to align the P proteins based on NTD/OD/XD separately, but the homogeneity/similarity was still very low. We therefore have now deleted the P protein alignment from the original figure 7b.

Figures 6 and 7: the NDV L/P complex is similar to VSV L/P complex in which the CTDs are on the opposite side of CD-MTase compared to that of PIV5 L/P complex. In Figure 6, the CD and MTase domains locate similarly at the top of the RdRp and PRNTase domains, while the CTD changes a lot between NDV/VSV and PIV5. In this paper, the author also claimed that the NDV L/P represents an intermediate state between VSV/hMPV and PIV5 L/P (line 228). However, there is no evidence showing that the CTD will rotate to the other side of CD-MTase during RNA synthesis in the same polymerase. Besides, the function of the intrusion loop conformation changes is unclear.

Our response: We thank the reviewer for the constructive questions. Previous studies suggested that the CTD is responsible for binding RNA and regulates the different methyltransferase activities, and different positions of CTD domain might reflect the switching on/off of specific MTase activities (J Virol. 2020 Jun 1;94(12):e00520-20.). However, no structure of nsNSV L protein with RNA has been reported to date, which is also the case for our NDV RNA polymerase. We speculate that when L protein is present in its capping state, the CTD is initially located at the gate of product exit channel to catch 5' of nascent RNA for its methylation (as shown in NDV). When the cap methylation is finished, it is speculated for CTD and MTase to rotate to the opposite position, away from the product exit channel for nascent RNA to leave RdRp-PRNTase region of L protein. Since no structural/experimental evidence is available, the precise role CTD plays in NDV RNA synthesis needs to be verified by future experiments, which are beyond the scope of this current work.

To date, very limited description has been made for function of the intrusion loop. When we compare the structures of NDV with other polymerases, we found that the intrusion loop showed wiggle change. Displacement of this intrusion loop might be required to accommodate RNA in the active site and regulate transcription initiation. During RNA transcription, the growing RNA may push the flexible intrusion loop with RNA bound to the conserved HR motif toward the active site of MTase-CTD. Unfortunately, currently a structure of RNA bound with nsNSV L protein is unavailable; therefore, it is impossible to testify the detailed function of the intrusion loop conformational change.

Minor comments:

The mapped part of the P protein in Figure 2a and Supplementary Figure 2a should be different.

Figure 3c, should they be 3' instead of 5'. And is the last one c3'-50 nts instead of 60 nts as the table 3.

Our response: We thank the reviewer for pointing the error. We have modified the mapped part of P protein in Supplementary Figure 2a (now as Supplementary Figure 3a in our revised manuscript) to match its length. In addition, we have modified the

former Fig. 3c (now as Figure 1b in our revised manuscript), in which 3' and c3'-50nt have been corrected.

Supplementary Figure 3: which RNA template was used in those figures? 40 nts? The bands were between 40 mer and 50 mer.

Our response: The template RNA used in this assay was c3'-40nt RNA. Presumably, the migration of the 40nt RNA product is located between 40-50mer markers for two reasons: 1) The actual molecular weight for the RNA product from v40 template is 13.21 kDa, while the molecular weight for 40mer and 50mer markers are 12.85 kDa and 16.02 kDa each; 2) the difference of loading volume for RNA product (10 μ L) and RNA marker (1 μ L).

Line 13: "Newcastle disease virus (NDV)" NDV abbreviation

Our response: We have added the abbreviation at line 13.

Line 43: 15.2k nts

Our response: We have added "s" at the end of "nt" at line 43.

Line 103: the concentration of the sample is 1mg/ml, while in line 342 it says 0.8 mg/ml. Which one is correct?

Our response: We thank the reviewer for pointing this out. The sample we used were 1 mg/ml, we have corrected line 342 to 1 mg/ml.

Line 192-205: Please specify Figures 5b, 5c, and 5d in corresponding locations, not only the last one. Figure 5e is not mentioned in the text.

Our response: We thank the reviewer for this suggestion. We have specified Figures 5b, 5c, and 5d in corresponding locations, and mentioned Figures 5e in the result section.

Line 280: PRNTase moves away??

Our response: We thank the reviewer pointing this question. We have rephased this sentence to "Subsequently, the movement of PRNTase opens the RNA product exit channel."

Line 387: Lf-Pe should be Lc-Pe

Our response: We thank the reviewer for pointing the mistake. We have changed Lf-Pe to Lc-Pe.

Line 395: NTPs (... , 124 nM GTP,887 nM GTP???) AND 165 nM of [α -32P]-ATP (3000 Ci/mmol). Two GTPs? And is the ATP 10 mCi/ml? Is any cold ATP added to the reaction?

Our response: We thank the reviewer for pointing the error. The "887 nM GTP" should be "887 μ M CTP". The radioactive ATP is indeed 10 mCi/ml. Cold ATP was added to the reaction mixture, at 33.36 μ M final concentration. We have now corrected the method of *in vitro* enzymatic assay.

Line 405-407: what are the sequences for the ladders? Are they the same as the products?

Our response: The sequences for the ladders have been included in Supplementary Table 3. We have designed the ladders as the terminal 3'-vRNA in equivalent lengths. The ladder sequences are not identical to the sequences of the RNA synthesis products.

REVIEWERS' COMMENTS

Reviewer #1 (Remarks to the Author):

The authors have satisfactorily addressed my previous comments; I have no further comments.

Reviewer #2 (Remarks to the Author):

The authors have adequately addressed my concerns in the revised manuscript. The new title is an improvement.

Reviewer #3 (Remarks to the Author):

I want to thank the authors for addressing the initial comments. Following the revision to the article, I do not have more questions now.